# Neural Translation of Input Specifications into Formal Grammars for Test Case Generation

## Abstract

Test cases are crucial for ensuring the program's correctness and evaluating performance in programming. The high diversity of test cases within constraints is necessary to distinguish between correct and incorrect answers. Automated source code generation is currently a popular area due to the inefficiency of manually generating test cases. Recent attempts involve generating conditional cases from problem descriptions using deep-learning models that learn from source code. However, this task requires a combination of complex skills such as extracting syntactic and logical constraints for a given test case from a problem, and generating test cases that satisfy the constraints. In this work, we introduce a modified context-free grammar that explicitly represents the syntactical and logical constraints embedded within programming problems. Our innovative framework for automated test case generation separates restriction extraction from test case generation, simplifying the task for the model. Our experimental results show that, compared to current methods, our framework produces test cases that are more precise and effective. All the codes in this paper are available in `https://anonymous.4open.science/r/neural_translation_for_test_case_generation`.

## 1 Introduction

Automated Test Case Generation (ATCG) is a growing area of interest within the field of software engineering, driven by the rapid progress of deep learning. These developments have led to the creation of numerous tools that enhance productivity in programming by offering source code suggestions through deep-learning models. While the ATCG plays a pivotal role in ensuring the accuracy of machine-generated codes, it is important to note that passing certain test cases does not guarantee the program's correctness. Recently, Liu et al. (2023) reported that there exist incorrect codes in the current program synthesis benchmarks that are not sufficiently verified due to the lack of test cases. To generate more test cases, they employed the famous large language model ChatGPT by OpenAI to produce the initial test cases by providing several prompts such as 'generate difficult inputs' or 'generate corner-case inputs' and applying type-aware mutations to the generated test cases.

We concentrate on the competitive programming field, which presents substantial demands for ATCG technologies for scoring solutions. In response to these demands, we introduce a neural translation task that converts natural language specifications into formal grammars. This approach allows the model to represent the meaning of the understood specifications without the need for further test case generation, enabling the model to focus on natural language understanding during the learning process. For the remaining aspects of test case generation, we rely on the formal grammar-based sampling algorithms. Our contribution extends to the introduction of context-free grammars with counters (CCFGs), meticulously designed to represent the syntactical and logical constraints that arise in competitive programming problem descriptions. We propose the use of a pre-trained CodeT5 model as a neural translation model to facilitate the neural translation task. Finally, we assess the utility of these approaches through experiments conducted on DeepMind's CodeContests dataset (Li et al., 2022), thereby validating the effectiveness of our CCFGs. Figure 1 describes our approach on the test case generation problem from specifications.

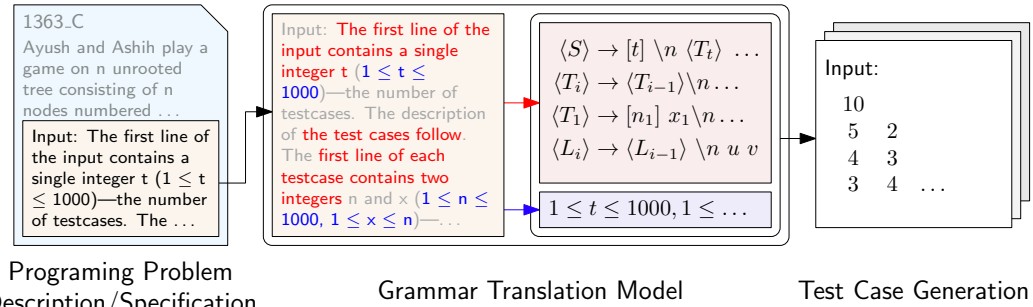

Figure 1: Overview of the proposed framework for generating test cases for competitive programming problems: The deep learning model translates specification into CCFGs while preserving their meaning. Subsequently, the CCFGs are utilized to generate test cases for the problem.

## 2 RELATED WORK

### 2.1 AUTOMATIC TEST CASE GENERATION

A review of the existing work done has indicated that the use of ATCG has shown significant improvement in the generation of Grammar-based Test Case Generation. Reports shows that the tools for automatically generating the test cases are becoming one of the common practices in large software organisations. The use of ATCG can enhance the efficiency and the ad-hoc in the software engineering field (Brunetto et al., 2021). But one of the key challenges is the navigation of the large input space and all the existing works struggle, or find difficulty in generating the high quality and well-structured test case (Olsthoorn, 2022). Typically, a deep learning approach takes an input specification as input and trains the model to get an accurate test case by using a neural network. Previously, A3Test uses the existing knowledge from an assertion generation task to the test case generation task (Alagarsamy et al., 2023). Similarly, Wang et al. (2022) uses the specifications from the natural language to generate the test cases by extracting the constraints and thus reduces the manual errors for generating the test cases.

### 2.2 NATURAL LANGUAGE TO FORMAL GRAMMAR

Many approaches have been made to convert the Natural Language to the Grammar. For example, Kate et al. (2005) implemented a method for inducing transformation rules that map natural-language sentences into a formal query or command language. Recently, research of Chen et al. (2023) shows that semantic regexes can better support complex data extraction tasks than standard regular expressions and the use of the regular expressions or had significantly outperformed the existing tools, including the state-of-the-art neural networks and program synthesis tools.Though, previous works also mentioned the drawback of using the regular expression as study shows that the conversion of the Natural Language to the regex fails to generate complex regexes. Ye et al. (2020) solved these issue by introducing the semantic parser that can be trained purely from the weak supervision based on the correctness of the synthesized regex. Similarly, Hahn et al. (2022) mentioned that the language models have the capability to translate the Natural Language to the formal specifications while maintaining the important keywords as well as outperforming the state of the art of using the regular expressions, without a particular need for domain-specific reasoning.

### 2.3 LARGE LANGUAGE MODELS FOR PROGRAM UNDERSTANDING AND GENERATION

There have been numerous studies on pre-training methods for understanding programming languages. Feng et al. (2020) proposed CodeBERT, which is a RoBERTa-based model pre-trained on multiple programming languages with masked language modeling. Guo et al. (2021) introduced GraphCodeBERT which is strengthened from CodeBERT by incorporating data flow information in the pre-training stage. Jiang et al. (2021) introduced TreeBERT, a tree-based pre-trained model that focuses on utilizing the extracted tree structure by encoding an abstract syntax tree as a set of composition paths. TreeBERT is trained by two novel objectives called tree-masked language mod-

eling and node order prediction. Rozière et al. (2021) investigated another programming-language-oriented pre-training objective based on the de-obfuscation of identifier names in source code.

Recently, Ahmad et al. (2021) proposed PLBART—Program and Language BART—which learns the interaction between program codes and natural language descriptions by leveraging the idea of denoising auto-encoder that uses a bidirectional encoder and an auto-regressive decoder. Yue Wang & Hoi (2021) introduced CodeT5, which leverages the code-specific characteristics in the pre-training stage by employing the new objectives such as masked random token prediction, masked identifier prediction, and identifier prediction objectives. Ma et al. (2021) proposed a pre-trained multilingual encoder-decoder model that regards the decoder as the task layer of off-the-shelf pre-trained encoders in order to take the advantage of both the large-scale monolingual data and bilingual data. Guo et al. (2022) demonstrated a unified cross-modal pre-trained model for programming language. This utilizes mask attention matrices with prefix adapters to control the behavior of the model and leverages cross-modal contents like an abstract syntax tree, and enhances the code representation by retaining all the structural information from the tree.

## 3 METHODOLOGY

In this section, we present a formal definition of *context-free grammars with counters*, specialized formal grammars designed to accurately represent the semantics of input specifications in the context of competitive programming. For symbols occur in each input specification, (1) *variables* are symbols used to represent values that vary depending on the test case, and (2) *terminals* are the symbols that are not variables, such as white spaces or newline symbols.

We often denote a set $X_I$ and one of its elements $x_i$, using a subscript $I$ and $i$. Then, we write the set $X_I \times A$ as $X_A$ and its element $(x_i, a) \in X_I \times A$ as $x_a \in X_A$ for simplicity. For example, if $X_I = \{x_i, y_i\}$ then $X_{\{0,1\}} = \{x_0, x_1, y_0, y_1\}$ and $x_0 = (x_i, 0)$.

$V$ and $T$ represents a sets of variables and terminals, respectively. We assume that we can divide $V$ into two disjoint sets: a set $V_\circ$ of *non-indexable variables*, and a set $V_I$ of *indexable variables*. For a problem input specification, we formalize its logical constraints as a set $\mathcal{A}$ of allowed variable assignments $\alpha : V_\circ \cup V_{\mathbb{N}_0} \to \mathbb{Z}_\perp$, where $\mathbb{Z}_\perp := \mathbb{Z} \cup \{\perp\}$. Also, we formalize its syntactical constraints as a function $\mathcal{L}$ maps an assignment $\alpha$ to a language $\mathcal{L}(\alpha) \in \Gamma^*$ on a *scheme alphabet* $\Gamma := T \cup V_\circ \cup V_{\mathbb{N}_0}$.

We introduce the *context-free grammar with counters* to represent $\mathcal{A}$ and $\mathcal{L}$.

**Definition 3.1.** *Let the* scheme alphabet $\Gamma := T \cup V_\circ \cup V_{\mathbb{N}_0}$. *A context-free grammar with counters is a tuple $G = (V, C, N, T, P, S, \mathcal{C})$, where (1) $C \in V_\circ$ is a set of* counter variables*; (2) a finite set of* nonterminals $N$ *is the disjoint union of a set $N_\circ$ of* indexable nonterminals $\langle \mathrm{X} \rangle$ *and a set $N_I$ of* non-indexable nonterminals $\langle \mathrm{Y_i} \rangle$*; (3) $P$ is a set of* productions *where $\Gamma' := \Gamma \cup N_\circ \cup N_{(\mathbb{N}_0 \cup C)} \cup \{[c] \mid c \in C\}$, and each production is an element of the following sets:*

1. $\{(x \to \gamma) \mid x \in N_\circ \cup N_{\mathbb{N}_0} \cup C, \gamma \in (\Gamma')^*\}$, *or*

2. $\{(\langle \mathrm{X_i} \rangle \to \gamma) \mid \langle \mathrm{X_i} \rangle \in N_I, \gamma \in (\overline{\Gamma'})^*\}$

*where $\overline{\Gamma'} := V_{\{i,i-1\}} \cup N_{\{i,i-1\}} \cup \Gamma'$; (4) $S \in N$ is a* start nonterminal*; and (5) $\mathcal{C}$ is a set of* constraints*, where each constraint is one of following forms:*

1. $(x \leq K)$ *or* $(K \neq x)$ *for* $x \in V_\circ \cup V_I \cup V_{\{\mathbb{N}_0 \cup C\}}$ *and* $K \in \mathbb{Z}$;

2. $(x \leq K)$ *or* $(K \neq x)$ *for* $x \in V_\circ \cup V_I \cup V_{\{\mathbb{N}_0 \cup C\}}$ *and* $K \in \mathbb{Z}$.

Let $\mathrm{idxing}_k(\gamma)$ for each $k \in \mathbb{N}_0$ be a string obtained by replacing subscript $i$'s and $i - 1$'s with $k$ and $k - 1$, respectively. Then we define a derivation $\vdash^*$ of $G$ as the following.

**Definition 3.2.** *For a CCFG $G$, a* derivation relation $\vdash^*_G$ *with respect to $G$ be the reflexive-transitive closure of $\vdash_G$, where $\vdash_G$ is a relation on $(\Gamma')^* \times \{\beta \mid \beta : C \to \mathbb{Z}_\perp\}$ defined as follows.*

*For $u, v, \gamma \in (\Gamma')^*$, $x \in \Gamma'$ and $\beta, \beta' : C \to \mathbb{Z}_\perp$ the relation $(uxv, \beta) \vdash_G (u\gamma v, \beta')$ holds if and only if they satisfies the following three conditions.*

- $\beta(c) \neq \bot$ *implies* $\beta'(c)$ *for every* $c \in C$,

- $\alpha'(c') \neq \bot$ *for every counter variable* $c'$ *occurs in* $\gamma$, *and*

- *one of the followings holds:*

    1. $x = \langle X \rangle \in N_\circ$, $(\langle X \rangle \rightarrow \gamma) \in P$;
    2. $x = \langle X_k \rangle \in N_{\mathbb{N}_0}$, $(\langle X_i \rangle \rightarrow \gamma') \in P$, *where* $\gamma' \in (\overline{\Gamma'})^*$ *with* $\mathrm{idxing}_k(\gamma') = \gamma$ *and* $(\langle X_k \rangle \rightarrow \gamma'') \notin P$ *for any* $\gamma'' \in (\Gamma')^*$;
    3. $x = \langle X_k \rangle \in N_{\mathbb{N}_0}$, $(\langle X_k \rangle \rightarrow \gamma) \in P$;
    4. $x = \langle X_c \rangle \in N_C$ *for some* $c \in C$, $\gamma = \langle X_{\alpha(c)} \rangle$; *or*
    5. $x = [c]$ *for some* $c \in C$, $([c] \rightarrow \gamma') \in P$ *with* $\gamma = c\gamma'$.

Finally, $G$ represents logical constraints $\mathcal{A}$ and syntactical constraints $\mathcal{L}$, where

- $\mathcal{L}(\alpha) = \{x \in \Gamma^* \mid (S, \alpha_0) \vdash_G^* (x, \alpha)\}$, for every $\alpha : C \rightarrow \mathbb{Z}$; and

- $\mathcal{A} := \{\alpha \mid \alpha : V_\circ \cup V_{\mathbb{N}_0} \rightarrow \mathbb{Z}_\bot \text{ satisfies all constraints in } \mathcal{C}\}$.

When labeling the dataset, we simplify the representation of $G$ with the following rules: (1) We present $P$ as a list `Productions` of productions $p \in P$. (2) We present $\mathcal{C}$ as a list `Constraints` of constraints and combination constraints such as "`a <= b < c`", which is a combination of "$a \leq b$, $b \leq c$ and $b \neq c$". (3) We do not present variables $V$, counter variables $C$, nonterminals $N$, and terminals $T$ explicitly. Instead, one can constructs them by analyzing `Productions` and `Constraints`. (4) We omit the representation of a start nonterminal $S$, by forcing the first production in `Productions` be the production of $S$.

### 3.1 String sampling and parsing of CCFGs

Remaining of this section, we assume that CCFGs and context-free grammars (CFGs) have no empty-productions that derive the empty string from nonterminals. Given an CFG $G$, one can sample a random string from $L(G)$ in linear time with respect to the length of the sampled string by simulating its derivation procedure probabilistically. If there are no constraints $(a \leq b)$ or $(a \neq b)$— that is, if every variables are independent—we can directly apply the sampling algorithm for CFGs to CCFGs, utilizing random assignment on variables during the derivation.

Otherwise, CCFGs have no polynomial time sampling algorithms due to the constraints; this is an immediate corollary of the co-NP-hardness of the emptiness problem.

**Theorem 3.1.** *For a given CCFG* $G = (V, C, N, T, P, S, \mathcal{C})$, *the emptiness problem of* $G$— *determining whether* $L(G) = \emptyset$ *or not—is co-NP-Hard.*

*Proof.* We use a reduction from the graph coloring problem (Karp, 1972). Let $(G, k)$ with $G = (V, E)$ be an instance of the graph coloring problem. Suppose that $V = \{v_1, v_2, \ldots, v_n\}$. Then we construct a CCFG $G := (V, \emptyset, \{\langle S \rangle\}, \emptyset, P, \langle S \rangle, \mathcal{C})$ where $P$ and $\mathcal{C}$ are defined as follows: (1) $P = \{\langle S \rangle \rightarrow v_1 \cdot v_2 \cdot \cdots \cdot v_n\}$ and (2) $\mathcal{C} = \{(0 \leq v \leq k) \mid v \in V\} \cup \{(u \neq v) \mid (u, v) \in E\}$. Then $G$ has $k$-coloring if and only if $L(G)$ is not empty. $\qquad \square$

We deal with the hardness of the sampling from CCFGs with straightforward Las Vegas approach— we sample variables until they satisfy the constraints.

One can utilize CCFGs as parser for test case validation according to their specifications. For parsing, we search derivations in depth-first way, and backtracks if the constraint or test case scheme violates the input string. Most of the grammars have linear parsing time with respect to the test-case length in real world, despite of the worst-case time complexity of the parsing algorithm being exponential. This is because most of the problem specifications are unambiguous, and so we can determine which derivation to choose for each step with 1-lookahead (Rosenkrantz & Stearns, 1969). Figure 2 describes a test case generation and validation using CCFGs.

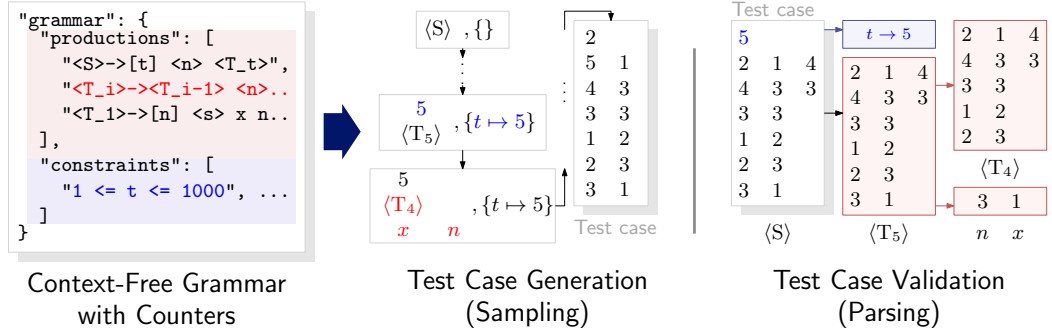

Figure 2: Test case generation and validation process using sampling and parsing algorithms of CCFGs.

## 4 EXPERIMENTS

We evaluate the practical usefulness of CCFGs by experiments. You can find all the implementations and codes used in experiments on Anonymous GitHub at `https://anonymous.4open.science/r/neural_translation_for_test_case_generation`.

### 4.1 IMPLEMENTATION DETAILS

**Specification extraction**   We heuristically extract the specification from the entire description, by finding the sections starts with "`Inputs:` " and "`Constraints:` ". If a description does not contain any of the two sections, we use the entire description as the specification.

**Context-free grammars with counters**   The CCFG described above aims to provide a proper formalization of the data used in the experiment. However, a discrepancy remains with our dataset. For instance, the production '` -> N  [a-z]{N}`' contains a regex-like expression that generates length $N$ upper-case alphabet string following the number $N$.

### 4.2 DATASET

We use CodeContests dataset, which consists with 3.76k train data, 97 valid data and 165 test data. We filter the data containing solutions written with Python3, since we use the Python3 solutions for evaluation. We exclude solutions that use `sys.setrecursionlimit`, since running such solutions by experimental code causes segmentation faults for edge case inputs and it makes operation system forcefully stops the experiments.

We manually labeled CCFGs for 700 and 136 problems from the training data in CodeContests, and use them for the model training and testing the syntactic equivalence of the grammar, respectively. On the other hand, during the model training, we use remaining unlabeled training data for pseudo labeling, and original 97 valid data for evaluation during development.

### 4.3 CCFGT5 MODEL ARCHITECTURE

We propose a translation model CCFGT5 that receives a problem input specification and produces a CCFG that has the same semantics. Our model utilizes two fine-tuned CodeT5 modules for generating CCFG components, namely one module for `Productions` and another for `Constraints`. For training, CCFG tokenizer translate each component as semicolon-separated list. The tokenizer add indicator words for each nonterminal and variable of productions, for example, it translates "`<X_i>`" to "`nonterminal X subscript i`" and "`a_i`" to "`variable a subscript i`". Then, the semicolon-separated list is converted into an integer array using RoBERTa tokenizer.

## 4.4 BASELINES

**CodeContests dataset and fine-tuned model** The CodeContests dataset provides a set of test cases for each problem in the dataset. Each test case falls into one of the categories: public, private and generated. We use the public dataset to fine-tune a pre-trained CodeT5-base model, which takes an algorithm code as input and produce a test case for the code. We use test cases in CodeContests dataset and test cases generated with the fine-tuned deep learning model for the comparison with grammar-based test cases.

**Mutation-based fuzzing** In algorithm programming problems, test cases often come with a specific input format provided at the end of the problem description. A typical example is where the first line specifies the number of upcoming inputs, and the second line contains integers equal to the number given in the first line, or a string of length equivalent to the integer provided in the first line. Hence, it is often convenient to keep spaces, string lengths, and newline characters the same while only changing the values of integers or characters constituting the strings to satisfy the input specification. By splitting the public test cases of CodeContests based on spaces and newline characters and then randomly sampling 30% of the tokens, and performing mutations based on the type of each token (e.g., integer, float, or string), one can achieve fuzzing while still satisfying the original input specification.

**Large language models** We use two famous large language models (LLMs), OpenAI's ChatGPT (in version 4.0) and Google's Bard. We uses the LLMs as either (1) baseline neural transition model from problem specifications to CCFGs, and (2) test case generation model for competitive programming. We test the performance of test case grammars generated by the LLMs by providing different numbers of examples 1 and 5. Full prompts for the each translation tasks are placed in Appendix A.

## 4.5 EVALUATION METRICS

To evaluate the model-translated CCFGs, we establish three distinct metrics: (1) *syntactic equivalence*, compare the model-translated CCFGs with human-labeled ones, (2) *soundness* and *completeness* compare the semantics of CCFGs with specifications, and (3) *effectiveness* measure the ability to distinguish correct and incorrect source codes.

**Syntactic equivalence** First, we compare the CCFGs translated by model with the ones labeled by humans. It should be note that CFGs are CCFGs without variables and constraints. Deciding the semantic equivalence of the CCFGs is undecidable as well as the equivalence of the CFGs (Hopcroft et al., 2007). Hence, we measure syntactic equivalence between models, which is a sufficient condition for semantic equivalence.

Although the names of nonterminals are unrelated to the semantics of the CCFGs, a naive comparison of productions would lead to evaluating CCFGs that differ only in the name of the nonterminals. Therefore, we normalize the presentation prior to comparing for equivalency by renaming nonterminals according to the order of their first occurrences in `Productions` in order to prevent unnecessary differentiation. Finally, we generate top-$k$ lists of productions and generations for each problem with our model. Then, we evaluate that exact match with human-label after the normalization.

**Soundness and completeness** The grammar must satisfy two properties for use in the test case generation: (1) the test case generated by the grammar has to be a valid input, and (2) the grammar should be able to generating every valid test case. We refer to these properties as 'soundness' and 'completeness' of the grammar, respectively.

We generate ten test cases and sample 10 correct solution codes from the dataset to evaluate the soundness. Due to mislabeled correct solutions in the dataset, sampled solutions may produce different outputs even if there exists a unique accurate solution. Consequently, we do not require every solution code to produce identical output for a test case. Instead, we define a test case to be *valid* if at least 80% of the solutions yield the same output. If all cases are valid, we consider the grammar to be sound. We ensure the completeness of a grammar by verifying that the grammar can parse 10 sampled public tests cases of the dataset.

### 4.5.1 EFFECTIVENESS

The primary purpose of test cases in competitive programming is to discern between correct and incorrect program codes. We evaluate the effectiveness of the generated test cases by assessing their ability to fulfill this objective.

For given sets $P$ and $N$ of correct and incorrect solution programs, we define the *effectiveness* $E(x)$ of a test case $x$ as

$$E(x) := |\{n \in N \mid n(x) \neq y^*\}|/|N|, \text{ where } y^* := \underset{y \in \{p(x)|p \in P\}}{\arg\max} |\{p \in P \mid p(x) = y\}|.$$

That is, $E(x)$ is the ratio of the distinguished incorrect solutions to the incorrect solutions when $y^*$ is the most probable correct output for $x$. We measure the ratio of the distinguished incorrect solutions, using 20 correct solutions and 20 incorrect solutions in the dataset. We generates 10 test cases using the grammar to assess this.

## 4.6 ANALYSIS OF EXPERIMENTAL RESULTS

In this section, when we refer to 'average,' it represents the overall average calculated from the individual averages for each problem. This approach ensures that our analysis remains unbiased by the varying number of test cases in each problem.

### 4.6.1 SYNTACTIC EQUIVALENCE

Table 1: Experimental results on syntactic equivalence. The values in parentheses represent measurements where cases were ignored. For CCFG, we generate 10 set of productions and 10 sets of constraints with with beam size 10. The accuracy for grammar was evaluated by probing $k^2$ grammars, which is constitutes combinations of $k$ productions and $k$ constraints.

| | | Exact Match Accuracy (%) | | |
| Model | Method | Productions | Constraints | Grammar |
|---|---|---|---|---|
| Bard | 1-shot | 9.56 | 55.15 | 9.56 |
| | 5-shot | 7.35 | 63.24 | 5.88 |
| ChatGPT | 1-shot | 7.35 | 59.56 | 5.88 |
| | 5-shot | 27.94 | 41.91 | 17.65 |
| CCFGT5 | Top-1 | 72.79 | 57.35 | 47.79 |
| | Top-5 | 80.88 | 69.85 | 62.50 |
| | Top-10 | 81.62 | 72.79 | 65.44 |

The experimental results in Table 1 clearly demonstrate that the CCFGT5 model outperforms LLMs in the task of translating specifications into grammars. This trend persists when only looking at the productions. Such results were expected as CCFGT5 has more opportunities to learn about CCFG owing to its reliance on 700 training data. Conversely, LLMs depend on 1-shot or 5-shot learning. As a result, fine-tuning is imperative for extracting syntax of valid problem inputs.

On the contrary, the results of the constraint analysis are unexpected. Large language models outperform fine-tuned models even in the 5-shot scenario. It is suggested that this may be attributed to the explicit mention of constraints in parentheses in the specifications. LLM detects this effortlessly with just five examples and uses the strings enclosed in the parentheses as constraints. Based on this analysis, it is anticipated that utilizing rule-based symbolic matching is a more effective method of extracting constraints compared to relying on deep learning-based approaches.

### 4.6.2 SEMANTIC EVALUATION

The evaluation results on Table 2 indicate that the CCFGT5 model, when used with a beam size of 10, generates the most semantically correct grammars. The difference in semantic between the model with a beam size of 100 and that with a beam size of 10 is not statistically significant. Note the relationship between beam size and the number of previous tokens considered during the generation.

Table 2: Overall comparison of evaluation metrics for semantics accuracy in generated grammars. Each column for semantic evaluation denotes the ratio of (1) Sound, (2) Complete, and (3) both Sound and Complete grammars to generated grammars.

| Model | Method | Semantic Evaluation (%) | | |
| --- | --- | --- | --- | --- |
| | | Soundness | Completeness | Sound. & Comple. |
| Bard | 1-shot | 12.50 | 15.44 | 11.76 |
| | 5-shot | 19.85 | 22.79 | 19.12 |
| ChatGPT | 1-shot | 13.24 | 12.50 | 11.03 |
| | 5-shot | 48.53 | 52.94 | 39.71 |
| CCFGT5 | Greed | 12.50 | 14.71 | 11.03 |
| | Beam(10) | **63.97** | **79.41** | **58.82** |
| | Beam(100) | **63.97** | 78.68 | **58.82** |

This observation suggests that tokens in the generated grammar heavily depend on those within a small range, approximately 10 tokens.

To delve deeper into this phenomenon, we conducted a statistical analysis of human-labeled train-grammar data. Our findings reveal that, on average, a single production consists of 15.53 tokens, while a constraint comprises 10.30 tokens. It supports that he number of tokens significantly affecting a single token does not exceed the length of a production or constraint. This insight implies that when generating a set of productions and constraints—for any model utilizing the tokenizer of CCFGT5—we expect that beam search sizes greater than 20 do not have a significant impact.

### 4.6.3 STATISTICS FOR GENERATED TEST CASES

Table 3 shows the statistics for test cases, generated by either baseline algorithms or CCFGs. The CCFGT5-generated test cases effectively identify invalid solutions. This is due to their greater length compared to the others. CCFGT5 for beam-size 10 (shortly, $CCFGT5_{10}$) have an average length of 4410.80, whereas those in the public, private, and generated test cases in CodeContests are only 19.10, 55.24, and 35.78.

As discussed in Section 3, if the variables are independent, CCFGs can generate each test case in linear time. This time-efficiency highlights the advantage for generating lengthy tests. Competitive programming scenarios often require longer test cases to effectively distinguish incorrect solutions, dealing with issues related to execution time. Consequently, CCFGs is especially valuable for use in competitive programming.

$CCFGT5_{10}$ demonstrates a high validity across entire generated test cases, compared to the fine-tuning, mutation and direct test case generation with Bard approaches. This observation suggests that machine-translated CCFGs lean towards not generating test cases when faced with uncertainties, rather than producing invalid ones. In situations where validating generated test cases becomes challenging, such when no correct solutions are available, using CCFGs is a fail-safe approach.

### 4.7 LIMITATIONS

While this study has deeply inspected the context-free grammars with counters and has analyzed various ways to use the grammar in automated test case generation, it is essential to clarify the limitations on our experiments.

As mentioned in Section 4.1, we use variants of CCFGs for human-labeling. Unless our implementation for generating and parsing on CCFG purposes full supports of the human-labeled grammars, there are some features that our implementation cannot support. Model generated grammars may contains those unsupported features, and it results in the underestimation of the soundness of completeness in Section 4.6.2. Also, there are problems with complicated constraints which is hard to represent with CCFGs. We summarize examples of such problems in Appendix B.

Table 3: Statistics of test cases for each problem are as follows: (1) Coverage, which denote the ratio of problems with at least one test case. (2) Validity, representing the average ratio of valid test cases to total test cases. (3) Ineffectiveness, indicating the average $(1 - \text{effectiveness})$ of cases. The right two columns represent statistics when invalid test cases are removed. The seven columns below represent statistics for grammar-generated test cases. Standard deviation is indicated in small font.

| | | Entire Test Case (%) | | | Valid Test Case (%) | |
|---|---|---|---|---|---|---|
| **Category** | **Method** | **Coverage** | **Valid.** | **Ineffective.** | **Coverage** | **Ineffective.** |
| | Public | 100.0 | 85.42 | 0.09 ±.48 | 88.97 | 0.06 ±.20 |
| Contests | Private | 52.21 | 79.09 | 0.05 ±.28 | 46.32 | **0.03** ±.27 |
| | Generated | 100.0 | **85.43** | **0.03** ±.23 | **97.06** | **0.03** ±.24 |
| CodeT5 | Fine-tuning | 100.0 | 54.19 | 0.18 ±.72 | 89.71 | 0.07 ±.51 |
| Fuzzing | Mutation | 100.0 | 70.77 | 0.15 ±.66 | 91.18 | **0.03** ±.20 |
| Bard | Zero-shot | 100.0 | 62.81 | **0.11** ±.54 | 82.35 | 0.09 ±.50 |
| ChatGPT | Zero-shot | 100.0 | **85.68** | 0.12 ±.59 | **94.12** | 0.05 ±.33 |
| | 1-shot | 22.06 | 57.33 | 0.68 ±2.13 | 13.97 | 0.82 ±2.43 |
| Bard | 5-shot | 53.68 | 36.85 | 0.08 ±.58 | 20.59 | **0.00** ±.00 |
| | 1-shot | 31.62 | 43.49 | 0.28 ±1.53 | 16.18 | 0.55 ±2.10 |
| ChatGPT | 5-shot | 71.32 | 70.62 | 0.18 ±.84 | 54.41 | 0.07 ±.58 |
| | Greed | 30.15 | 42.93 | **0.01** ±.08 | 13.24 | **0.00** ±.00 |
| CCFGT5 | Beam(10) | **88.24** | **75.75** | 0.05 ±.46 | **72.79** | **0.00** ±.00 |
| | Beam(100) | **88.24** | 75.58 | 0.05 ±.46 | 72.06 | **0.00** ±.00 |

We assume that each problem has a unique value for a test case when measure the validness of the test case. In real, a few number of problem allows multiple outputs. It results in the under-estimation of the number of valid test cases during the experiments.

Our CCFG parsing algorithm exhibits a worst-case complexity that is exponential with respect to the length of the input string. In comparison, the CYK-algorithm, published by Sakai (1961), offers a polynomial-time solution for CFG parsing. It still remains an open problem, that there is their any polynomial-time CCFG parsing algorithm.

One notable limitation of our study pertains to the size of the training dataset, which may hinder the model's ability to generalize effectively. The design and implementation of grammar-based pseudo-labeling techniques for model generalization represent a promising avenue for our future research endeavors.

## 5 CONCLUSIONS

In this study, we present an effective approach for automated generation from descriptions commonly found in competitive programming tasks. Our methodology leverages formal grammar, introducing a context-free grammar with counters designed to accurately represent the meaning of input specifications. The grammar enables rapid test case generation, encompassing all valid cases, and allows for linear-time test case verification through parsing, for most CCFGs of competitive programming problems.

Our experiments demonstrate that our proposed CCFGT5 model outperforms few-shot generation approaches using large language models. The time-efficiency of the CCFGs sampling algorithm contributes to the effectiveness of the generated test cases in distinguishing incorrect solutions. Importantly, our use of CCFGs as a test case generation method proves to be a fail-safe option, especially in situations where validating inputs becomes challenging

In our future work, we plan to implement a robust pseudo-labeling framework for CCFGT5 model, with the aim of enhancing the semantic accuracy of the resulting CCFGs. Additionally, we intend to refine the current string sampling algorithm by incorporating weighted production rules from weighted CFGs (Salomaa, 1969). This enhancement will enable us to generate even more effective test cases from the grammar.

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

## A    PROMPTS FOR GRAMMAR GENERATION OF LARGE LANGUAGE MODELS

For the prompt of a few-shot grammar generation task with large language models, we choose a grammar with the moderate level to cover from simple specifications to complicated ones. Here we attach entire prompts for the few-shot generation. The string '{{specification}}' within the prompts is replaced with each specification of the problem.

### A.1    PROMPT FOR TRANSLATION INTO GRAMMAR: 1-SHOT

```
<Specification> "Input\n\nThe first line contains integer n (1 ≤ n ≤
↪  5·10^5), showing how many numbers are in the array. The second line
↪  contains n integers a[1], a[2], ..., a[n] (|a[i]| ≤ 10^9) | the
↪  elements of array a".</Specification>
<Grammar> " -> [N] <n> <T_N>", "<T_i> -> <T_i-1>  a_i", "<T_1> ->
↪  a_1" </Grammar>
<Constraint> "1<=n<=5*10^5" , "1<=a_i<=10^9" </Constraint>
<Specification> {{specification}} </Specification>
<Grammar>
<Constraint>
Generate the Grammar and the Constraint for the last specification
↪  similar to the examples provided
```

## A.2 PROMPT FOR TRANSLATION INTO GRAMMAR: 5-SHOT

```
<Specification> "Input\n\nThe first line contains integer n (1 ≤ n ≤
↪  5·10^5), showing how many numbers are in the array. The second line
↪  contains n integers a[1], a[2], ..., a[n] (|a[i]| ≤ 10^9) | the
↪  elements of array a".</Specification>
<Grammar> " -> [N] <n> <T_N>", "<T_i> -> <T_i-1>  a_i", "<T_1> ->
↪  a_1" </Grammar>
<Constraint> "1<=n<=5*10^5" , "1<=a_i<=10^9" </Constraint>
<Specification> "Constraints\n\n* 4 ≤ |S| ≤ 10 (|S| is the length of the
↪  string S.)\n* Each character of S is uppercase or lowercase English
↪  letter.\n\nInput\n\nInput is given from Standard Input in the
↪  following format:\n\n\nS" </Specification>
<Grammar> "->[a-zA-Z]{4,10}" </Grammar>
<Constraint> "" </Constraint>
<Specification> "Input\n\nThe first line contains one integer t\\ (1 ≤
↪  t≤ 100): the number of queries.\n\nEach query contains two lines.
↪  The first line contains one integer n\\ (1≤ n≤ 100 000): the number
↪  of models in the shop, and the second line contains n integers
↪  s_1,...,s_n\\ (1≤ s_i≤ 10^9): the sizes of models.\n\nIt is
↪  guaranteed that the total sum of n is at most 100 000."
↪  </Specification>
<Grammar> "->[t] <n> <T_t>", "<T_i>-><T_i-1> <n> [n] <n> <L_n>",
↪  "<T_1>->[n] <n> <L_n>", "<L_i>-><L_i-1>  s_i", "<L_1>->s_1"
↪  </Grammar>
<Constraint> "1<=t<=100", "1<=n<=100000", "1<=s_i<=10^9" </Constraint>
<Specification> "Input\n\nThe only line of the input contains four
↪  integers a, b, c, d (1 ≤ a, b, c, d ≤ 10^9). It is possible that any
↪  two (or all three) ropewalkers are in the same position at the
↪  beginning of the performance." </Specification>
<Grammar> "->a  b  c  d" </Grammar>
<Constraint> "1<=a<=10^9", "1<=b<=10^9", "1<=c<=10^9", "1<=d<=10^9"
↪  </Constraint>
<Specification> "Constraints\n\n* 1 ≤ N ≤ 10^5\n* 1 ≤ a_i ≤ 10^{9}\n*
↪  a_i are integers.\n\nInput\n\nInput is given from Standard Input in
↪  the following format:\n\n\nN\na_1 a_2 ... a_{3N}" </Specification>
<Grammar> "->[N] <n> <T_3N>", "<T_i>-><T_i-1>  a_i", "<T_1>->a_1"
↪  </Grammar>
<Constraint> "1<=N<=10^5", "1<=a_i<=10^9" </Constraint>
<Specification> {{specification}} </Specification>
<Grammar>
<Constraint>
Generate the Grammar and the Constraint for the last specification
↪  similar to the examples provided
```

## A.3 PROMPT FOR TEST CASE GENERATION

```
Generate 10 valid test cases for the following specification:
{{specification}}
Each test cases should be in one line using "\n"
```

## B A LIST OF DIFFICULT PROBLEMS TO WRITE GRAMMARS

| Problem ID | 977_A codeforces Wrong Subtraction |
| --- | --- |

| | |
|---|---|
| **Description** | Little girl Tanya is learning how to decrease a number by one, but she does it wrong with a number consisting of two or more digits. Tanya subtracts one from a number by the following algorithm:

• if the last digit of the number is non-zero, she decreases the number by one;

• if the last digit of the number is zero, she divides the number by 10 (i.e. removes the last digit).

You are given an integer number $n$. Tanya will subtract one from it $k$ times. Your task is to print the result after all $k$ subtractions.
It is guaranteed that the result will be a positive integer number.
Input
The first line of the input contains two integer numbers n and k ($2 \leq n \leq 10^9, 1 \leq k \leq 50$) — the number from which Tanya will subtract and the number of subtractions correspondingly. |
| **Note** | It is difficult to specify the condition that the result after $k$ subtractions in the input should be positive. |
| **Problem ID** | p03471 AtCoder Beginner Contest 085 - Otoshidama |
| **Description** | The commonly used bills in Japan are 10000-yen, 5000-yen and 1000-yen bills. Below, the word "bill" refers to only these.
According to Aohashi, he received an otoshidama (New Year money gift) envelope from his grandfather that contained N bills for a total of Y yen, but he may be lying. Determine whether such a situation is possible, and if it is, find a possible set of bills contained in the envelope. Assume that his grandfather is rich enough, and the envelope was large enough.
Constraints

• $1 \leq N \leq 2000$

• $1000 \leq Y \leq 2 \times 10^7$

• $N$ is an integer.

• $Y$ is a multiple of 1000.

Input
Input is given from Standard Input in the following format:
N Y |
| **Note** | Y is multiple 1000 |
| **Problem ID** | p03244 AtCoder Beginner Contest 111 - /\/\/\/ |
| **Description** | A sequence $a_1, a_2, \ldots, a_n$ is said to be / when the following conditions are satisfied:

• For each $i = 1, 2, ..., n - 2, a_i = a_{i+2}$.

• Exactly two different numbers appear in the sequence.

You are given a sequence $v_1, v_2, \ldots, v_n$ whose length is even. We would like to make this sequence / by replacing some of its elements. Find the minimum number of elements that need to be replaced.
Constraints

• $2 \leq n \leq 10^5$

• $n$ is even.

• $1 \leq v_i \leq 10^5$

• $v_i$ is an integer.

Input
Input is given from Standard Input in the following format:
n
$v_1\ v_2\ \ldots\ v_n$ |
| **Note** | n is even |
| **Problem ID** | p02682 AtCoder Beginner Contest 167 - Easy Linear Programming |

| Description | We have A cards, each of which has an integer 1 written on it. Similarly, we also have B cards with 0s and C cards with -1s.
We will pick up K among these cards. What is the maximum possible sum of the numbers written on the cards chosen?
Constraints

    • All values in input are integers.
    • $0 \le A, B, C$
    • $1 \le K \le A + B + C \le 2 \times 10^9$

Input
Input is given from Standard Input in the following format:
A B C K |
|---|---|
| **Note** | $0 \le A, B, C,\ 1 \le K \le A + B + C \le 2 \times 10^9$
The range of A, B and C is not clearly specified |
| **Problem ID** | p02730 AtCoder Beginner Contest 159 - String Palindrome |
| **Description** | A string S of an odd length is said to be a strong palindrome if and only if all of the following conditions are satisfied:

    • $S$ is a palindrome.
    • Let $N$ be the length of $S$. The string formed by the 1-st through $((N - 1)/2)$-th characters of $S$ is a palindrome.
    • The string consisting of the $(N + 3)/2$-st through $N$-th characters of $S$ is a palindrome.

Determine whether S is a strong palindrome.
Constraints

    • $S$ consists of lowercase English letters.
    • The length of $S$ is an odd number between 3 and 99 (inclusive).

Input
Input is given from Standard Input in the following format:
S |
| **Note** | The length of $S$ is an odd number |
| **Problem ID** | 25_A codeforces IQ test |
| **Description** | Bob is preparing to pass IQ test. The most frequent task in this test is to find out which one of the given n numbers differs from the others. Bob observed that one number usually differs from the others in evenness. Help Bob — to check his answers, he needs a program that among the given n numbers finds one that is different in evenness.
Input
The first line contains integer $n(3 \le n \le 100)$ — the amount of numbers in the task. The second line contains n space-separated natural numbers, not exceeding 100. It is guaranteed, that exactly one of these numbers differs from the others in evenness. |
| **Note** | Exactly one of these numbers differs from the others in evenness |
| **Problem ID** | p02707 AtCoder Beginner Contest 163 - management |

| Description | A company has N members, who are assigned ID numbers 1, ..., N. Every member, except the member numbered 1, has exactly one immediate boss with a smaller ID number. When a person X is the immediate boss of a person Y, the person Y is said to be an immediate subordinate of the person X. You are given the information that the immediate boss of the member numbered $i$ is the member numbered $A_i$. For each member, find how many immediate subordinates it has. Constraints

• $2 \leq N \leq 2 \times 10^5$

• $1 \leq A_i < i$

Input
Input is given from Standard Input in the following format:
N
$A_2 \ldots A_N$ |
|---|---|
| **Note** | $A_i < i$ |
| **Problem ID** | p03017 AtCoder Grand Contest 034 - Kenken Race |
| **Description** | There are N squares arranged in a row, numbered 1, 2, ..., N from left to right. You are given a string S of length N consisting of '.' and '#'. If the i-th character of S is '#', Square i contains a rock; if the i-th character of S is '.', Square i is empty.
In the beginning, Snuke stands on Square A, and Fnuke stands on Square B.
You can repeat the following operation any number of times:
* Choose Snuke or Fnuke, and make him jump one or two squares to the right. The destination must be one of the squares, and it must not contain a rock or the other person.
You want to repeat this operation so that Snuke will stand on Square C and Fnuke will stand on Square D.
Determine whether this is possible.
Constraints

• $4 \leq N \leq 200\,000$

• $S$ is a string of length $N$ consisting of '.' and '#'.

• $1 \leq A, B, C, D \leq N$

• Square $A, B, C$ and $D$ do not contain a rock.

• $A, B, C$ and $D$ are all different.

• $A < B$

• $A < C$

• $B < D$

Input
Input is given from Standard Input in the following format:
N A B C D
S |
| **Note** | $P_A, P_B, P_C, P_D \neq$ '#' |
| **Problem ID** | p02682 AtCoder Beginner Contest 167 - Easy Linear Programming |
| **Description** | We have A cards, each of which has an integer 1 written on it. Similarly, we also have B cards with 0s and C cards with -1s.
We will pick up K among these cards. What is the maximum possible sum of the numbers written on the cards chosen?
Constraints

• All values in input are integers.

• $0 \leq A, B, C$

• $1 \leq K \leq A + B + C \leq 2 \times 10^9$

Input
Input is given from Standard Input in the following format:
A B C K |
| **Note** | $K \leq A + B + C$
Maximum values of A, B, and C are undefined |

| Problem ID | p03282 AtCoder Beginner Contest 106 - To Infinity |
|---|---|
| Description | Mr. Infinity has a string S consisting of digits from '1' to '9'. Each time the date changes, this string changes as follows: |
| | • Each occurrence of '2' in S is replaced with '22'. Similarly, each '3' becomes '333', '4' becomes '4444', '5' becomes '55555', '6' becomes '666666', '7' becomes '7777777', '8' becomes '88888888' and '9' becomes '999999999'. '1' remains as '1'. |
| | For example, if S is '1324', it becomes '1333224444' the next day, and it becomes '13333333332222444444444444444444' the day after next. You are interested in what the string looks like after $5 * 10^{15}$ days. What is the K-th character from the left in the string after $5 * 10^{15}$ days? |
| | Constraints |
| | • $S$ is a string of length between 1 and 100 (inclusive). |
| | • $K$ is an integer between 1 and $10^{18}$ (inclusive). |
| | • The length of the string after $5 * 10^{15}$ days is at least $K$. |
| | Input |
| | Input is given from Standard Input in the following format: |
| | S |
| | K |
| Note | Constraints of K are associated with output |
| **Problem ID** | 1154_A codeforces Restoring Three Numbers |
| Description | Polycarp has guessed three positive integers a, b and c. He keeps these numbers in secret, but he writes down four numbers on a board in arbitrary order — their pairwise sums (three numbers) and sum of all three numbers (one number). So, there are four numbers on a board in random order: a+b, a+c, b+c and a+b+c. |
| | You have to guess three numbers a, b and c using given numbers. Print three guessed integers in any order. |
| | Pay attention that some given numbers a, b and c can be equal (it is also possible that $a = b = c$). |
| | Input |
| | The only line of the input contains four positive integers $x_1, x_2, x_3, x_4 (2 \leq x_i \leq 10^9)$ — numbers written on a board in random order. It is guaranteed that the answer exists for the given number $x_1, x_2, x_3, x_4$. |
| | Output |
| | Print such positive integers a, b and c that four numbers written on a board are values a+b, a+c, b+c and a+b+c written in some order. Print a, b and c in any order. If there are several answers, you can print any. It is guaranteed that the answer exists. |
| Note | Constraints are associated with output |
| **Problem ID** | p03014 AtCoder Beginner Contest 129 - Lamp |

| Description | There is a grid with H horizontal rows and W vertical columns, and there are obstacles on some of the squares. |
|---|---|
| | Snuke is going to choose one of the squares not occupied by an obstacle and place a lamp on it. The lamp placed on the square will emit straight beams of light in four cardinal directions: up, down, left, and right. In each direction, the beam will continue traveling until it hits a square occupied by an obstacle or it hits the border of the grid. It will light all the squares on the way, including the square on which the lamp is placed, but not the square occupied by an obstacle. |
| | Snuke wants to maximize the number of squares lighted by the lamp. |
| | You are given H strings $S_i$ $(1 \leq i \leq H)$, each of length W. If the j-th character $(1 \leq j \leq W)$ of $S_i$ is '#', there is an obstacle on the square at the i-th row from the top and the j-th column from the left; if that character is '.', there is no obstacle on that square. |
| | Find the maximum possible number of squares lighted by the lamp. |
| | Constraints |
| | • $1 \leq H \leq 2,000$ |
| | • $1 \leq W \leq 2,000$ |
| | • $S_i$ is a string of length W consisting of '#' and '.'. |
| | • '.' occurs at least once in one of the strings $S_i (1 \leq i \leq H)$. |
| | Input |
| | Input is given from Standard Input in the following format: |
| | H W |
| | $S_1$ |
| | $\vdots$ |
| | $S_H$ |
| Note | '.' occurs at least once in one of the strings |
| Problem ID | 1343_A codeforces Candies |
| Description | Recently Vova found n candy wrappers. He remembers that he bought $x$ candies during the first day, $2x$ candies during the second day, $4x$ candies during the third day, ..., $2^{k-1}x$ candies during the $k$-th day. But there is an issue: Vova remembers neither $x$ nor $k$ but he is sure that $x$ and $k$ are positive integers and $k > 1$. |
| | Vova will be satisfied if you tell him any positive integer $x$ so there is an integer $k > 1$ that $x + 2x + 4x + ... + 2^{k-1}x = n$. It is guaranteed that at least one solution exists. Note that $k > 1$. |
| | You have to answer $t$ independent test cases. |
| | Input |
| | The first line of the input contains one integer $t(1 \leq t \leq 10^4)$ — the number of test cases. Then $t$ test cases follow. |
| | The only line of the test case contains one integer $n(3 \leq n \leq 10^9)$ — the number of candy wrappers Vova found. It is guaranteed that there is some positive integer $x$ and integer $k > 1$ that $x + 2x + 4x + ... + 2^{k-1}x = n$. |
| Note | Must generate $n$ with $n = x + 2x + 4x + ... + 2^{k-1}x(k > 1)$ for some positive int $x$ |
| Problem ID | 1385_B codeforces Restore the Permutation by Merger. |

| Description | A permutation of length $n$ is a sequence of integers from 1 to $n$ of length $n$ containing each number exactly once. For example, $[1], [4, 3, 5, 1, 2], [3, 2, 1]$ are permutations, and $[1, 1], [0, 1], [2, 2, 1, 4]$ are not. |
| --- | --- |
| | There was a permutation $p[1...n]$. It was merged with itself. In other words, let's take two instances of $p$ and insert elements of the second p into the first maintaining relative order of elements. The result is a sequence of the length $2n$. |
| | For example, if $p = [3, 1, 2]$ some possible results are: $[3, 1, 2, 3, 1, 2], [3, 3, 1, 1, 2, 2], [3, 1, 3, 1, 2, 2]$. The following sequences are not possible results of a merging: $[1, 3, 2, 1, 2, 3], [3, 1, 2, 3, 2, 1], [3, 3, 1, 2, 2, 1]$. For example, if $p = [2, 1]$ the possible results are: $[2, 2, 1, 1], [2, 1, 2, 1]$. The following sequences are not possible results of a merging: $[1, 1, 2, 2], [2, 1, 1, 2], [1, 2, 2, 1]$. |
| | Your task is to restore the permutation $p$ by the given resulting sequence a. It is guaranteed that the answer exists and is unique. |
| | You have to answer $t$ independent test cases. |
| | Input |
| | The first line of the input contains one integer $t(1 \leq t \leq 400)$—the number of test cases. Then $t$ test cases follow. |
| | The first line of the test case contains one integer $n(1 \leq n \leq 50)$—the length of permutation. The second line of the test case contains $2n$ integers $a_1, a_2, ..., a_{2n}(1 \leq a_i \leq n)$, where $a_i$ is the $i$-th element of $a$. It is guaranteed that the array a represents the result of merging of some permutation $p$ with the same permutation $p$. |
| Note | It is guaranteed that the array a represents the result of merging of some permutation $p$ with the same permutation $p$. |
| **Problem ID** | p03033 AtCoder Beginner Contest 128 - Roadwork |
| Description | There is an infinitely long street that runs west to east, which we consider as a number line. |
| | There are N roadworks scheduled on this street. The $i$-th roadwork blocks the point at coordinate $X_i$ from time $S_i$ - 0.5 to time $T_i - 0.5$. |
| | $Q$ people are standing at coordinate 0. The $i$-th person will start the coordinate 0 at time $D_i$, continue to walk with speed 1 in the positive direction and stop walking when reaching a blocked point. |
| | Find the distance each of the Q people will walk. |
| | Constraints |
| | • All values in input are integers. |
| | • $1 \leq N, Q \leq 2 \times 10^5$ |
| | • $0 \leq S_i < T_i \leq 10^9$ |
| | • $1 \leq X_i \leq 10^9$ |
| | • $0 \leq D_1 < D_2 < ... < D_Q \leq 10^9$ |
| | • If $i \neq j$ and $X_i = X_j$, the intervals $[S_i, T_i)$ and $[S_j, T_j)$ do not overlap. |
| | Input |
| | Input is given from Standard Input in the following format: |
| | $N \ Q$ |
| | $S_1 \ T_1 \ X_1$ |
| | $\vdots$ |
| | $S_N \ T_N \ X_N$ |
| | $D_1$ |
| | $\vdots$ |
| | $D_Q$ |
| Note | * If $i \neq j$ and $X_i = X_j$, the intervals $[S_i, T_i)$ and $[S_j, T_j)$ do not overlap. |
| **Problem ID** | p02681 AtCoder Beginner Contest 167 - Registration |

| | |
|---|---|
| **Description** | Takahashi wants to be a member of some web service.
He tried to register himself with the ID $S$, which turned out to be already used by another user.
Thus, he decides to register using a string obtained by appending one character at the end of $S$ as his ID.
He is now trying to register with the ID $T$. Determine whether this string satisfies the property above.
Constraints

• $S$ and $T$ are strings consisting of lowercase English letters.
• $1 \leq |S| \leq 10$
• $|T| = |S| + 1$

Input
Input is given from Standard Input in the following format:
$S\ T$ |
| **Note** | $T = S + 1$ |
| **Problem ID** | 1374_C codeforce Move Brackets |
| **Description** | You are given a bracket sequence $s$ of length $n$, where $n$ is even (divisible by two). The string $s$ consists of $n/2$ opening brackets '(' and $n/2$ closing brackets ')'.
In one move, you can choose exactly one bracket and move it to the beginning of the string or to the end of the string (i.e. you choose some index $i$, remove the $i$-th character of $s$ and insert it before or after all remaining characters of $s$).
Your task is to find the minimum number of moves required to obtain regular bracket sequence from $s$. It can be proved that the answer always exists under the given constraints.
Recall what the regular bracket sequence is:

• "()" is regular bracket sequence;
• if s is regular bracket sequence then "(" + $s$ + ")" is regular bracket sequence;
• if s and t are regular bracket sequences then $s + t$ is regular bracket sequence.

For example, "()()", "(())()", "(())" and "()" are regular bracket sequences, but ")(", "()(" and ")))" are not.
You have to answer $t$ independent test cases.
Input
The first line of the input contains one integer t ($1 \leq t \leq 2000$) — the number of test cases. Then $t$ test cases follow.
The first line of the test case contains one integer $n$ ($2 \leq n \leq 50$) — the length of $s$. It is guaranteed that $n$ is even. The second line of the test case containing the string $s$ consisting of $n/2$ opening and $n/2$ closing brackets. |
| **Note** | It is guaranteed that $n/2$ is open parentheses and $n/2$ is closed parentheses |
| **Problem ID** | 1220_A codeforces Cards |
| **Description** | When Serezha was three years old, he was given a set of cards with letters for his birthday. They were arranged into words in the way which formed the boy's mother favorite number in binary notation. Serezha started playing with them immediately and shuffled them because he wasn't yet able to read. His father decided to rearrange them. Help him restore the original number, on condition that it was the maximum possible one.
Input
The first line contains a single integer n ($1 \leq n \leq 10^5$) — the length of the string. The second line contains a string consisting of English lowercase letters: 'z', 'e', 'r', 'o' and 'n'.
It is guaranteed that it is possible to rearrange the letters in such a way that they form a sequence of words, each being either "zero" which corresponds to the digit 0 or "one" which corresponds to the digit 1. |
| **Note** | It is guaranteed that it is possible to rearrange the letters in such a way that they form a sequence of words, each being either "zero" or "one" |
| **Problem ID** | p03290 AtCoder Beginner Contest 104 - All Green |

| Description | A programming competition site AtCode provides algorithmic problems. Each problem is allocated a score based on its difficulty. Currently, for each integer i between 1 and D (inclusive), there are $p_i$ problems with a score of 100i points. These $p_1 + \ldots + p_D$ problems are all of the problems available on AtCode.
A user of AtCode has a value called total score. The total score of a user is the sum of the following two elements:
* Base score: the sum of the scores of all problems solved by the user. * Perfect bonuses: when a user solves all problems with a score of 100i points, he/she earns the perfect bonus of $c_i$ points, aside from the base score ($1 \le i \le D$).
Takahashi, who is the new user of AtCode, has not solved any problem. His objective is to have a total score of G or more points. At least how many problems does he need to solve for this objective?
Constraints

• $1 \le D \le 10$
• $1 \le p_i \le 100$
• $100 \le c_i \le 10^6$
• $100 \le G$
• All values in input are integers.
• $c_i$ and G are all multiples of 100.
• It is possible to have a total score of G or more points.

Input
Input is given from Standard Input in the following format:
$D\ G$
$p_1\ c_1$
$\vdots$
$p_D\ c_D$ |
|---|---|
| Note | here are only two constructs for $G$: $100 \le G$, G are all multiples of 100.
And the maximum value of $G$ is not determined. |
| **Problem ID** | p02697 AtCoder Beginner Contest 165 - Rotation Matching |
| Description | You are going to hold a competition of one-to-one game called AtCoder Janken. (Janken is the Japanese name for Rock-paper-scissors.) $N$ players will participate in this competition, and they are given distinct integers from 1 through $N$. The arena has $M$ playing fields for two players. You need to assign each playing field two distinct integers between 1 and $N$ (inclusive). You cannot assign the same integer to multiple playing fields. The competition consists of $N$ rounds, each of which proceeds as follows:

• For each player, if there is a playing field that is assigned the player's integer, the player goes to that field and fight the other player who comes there.
• Then, each player adds 1 to its integer. If it becomes $N + 1$, change it to 1.

You want to ensure that no player fights the same opponent more than once during the N rounds. Print an assignment of integers to the playing fields satisfying this condition. It can be proved that such an assignment always exists under the constraints given.
Constraints

• $1 \le M$
• $M \times 2 + 1 \le N \le 200000$

Input
Input is given from Standard Input in the following format:
$N\ M$ |
| Note | $1 \le M$, $M \times 2 + 1 \le N \le 200000$
N comes before $M$, but $M$ must be defined in order to produce $N$ |
| **Extra Problems** | 977_A codeforces Wrong Subtraction
p03244 AtCoder Beginner Contest 111 - /
p03471 AtCoder Beginner Contest 085 - Otoshidama |

p02682 AtCoder Beginner Contest 167 - Easy Linear Programming
p02730 AtCoder Beginner Contest 159 - String Palindrome
25_A codeforces IQ test
p02707 AtCoder Beginner Contest 163 - management
p03017 AtCoder Grand Contest 034 - Kenken Race
p02682 AtCoder Beginner Contest 167 - Easy Linear Programming
p03282 AtCoder Beginner Contest 106 - To Infinity
1154_A codeforces Restoring Three Numbers
p03014 AtCoder Beginner Contest 129 - Lamp
1343_A codeforces Candies
1385_B codeforces Restore the Permutation by Merger.
p03033 AtCoder Beginner Contest 128 - Roadwork
p02681 AtCoder Beginner Contest 167 - Registration
1374_C codeforce Move Brackets
1220_A codeforces Cards
p03290 AtCoder Beginner Contest 104 - All Green
p02697 AtCoder Beginner Contest 165 - Rotation Matching

