# OpenReview forum: "Neural Translation of Input Specifications into Formal Grammars for Test Case Generation"
_ICLR.cc/2024/Conference — ICLR 2024 Conference Withdrawn Submission_

### Official Review · Reviewer_bjCm · 2023-10-30

**Soundness:** 2 fair
**Presentation:** 2 fair
**Contribution:** 2 fair
**Rating:** 3
**Confidence:** 2

**Summary:**

Test cases are crucial for determining whether programs are correct and how well they perform. However, for automated source code generation, it can be difficult to extract syntactic and logical constraints from problem descriptions. In this work, a context-free grammar, CCFG, that reflects these restrictions is introduced. In comparison to existing approaches, this novel framework makes the task simpler and generates test cases that are more accurate and effective.

**Strengths:**

+ The paper is on an important direction, proposing a modified context-free grammar for automated test case generation
+ The paper is with well-structured review of relevant literature

**Weaknesses:**

- The manuscript lacks of clarity in exposition and there are many details require clarification
- The motivation of designing CCFG and the steps in Section 3 is unclear
- The presentation of the paper is not good

**Questions:**

This work introduces a modified context-free grammar for automated test case generation, simplifying the task and producing more precise and effective test cases compared to current methods, enhancing the effectiveness of ATCG. The presentation of the paper is not good and makes the paper hard to follow. For example, the advantage of using the newly proposed grammar is not deeply discussed. In Section 3, the authors directly jump into the definition of CCFG, and the usage and motivation of the definitions are vague. Moreover, the experiment also misses many details and is not easy to read. I list the questions below.

1. Section 2: The authors could discuss more the automated test case generation without using grammar.

2. Definition 3.1, 3.2, Theorem 3.1: Why do you propose these definitions and theorems? A bit more motivation would be easier for readers to understand.

3. Section 4.1: What are these implementation details? Are they for CCFG? It seems like they are not that matters.

4.1 Section 4.2: You should provide the citation for the CodeContest dataset.
4.2 Section 4.2:
> We manually labeled CCFGs for 700 and 136 problems from the training data in CodeContests, ...

Why there are two different numbers of problems?

5. Section 4.3: The description of CCFGT5 architecture is too vague.

---

### Official Review · Reviewer_n1Av · 2023-10-31

**Soundness:** 2 fair
**Presentation:** 2 fair
**Contribution:** 2 fair
**Rating:** 3
**Confidence:** 3

**Summary:**

This paper addresses automated test case generation with the goal of automated grading of coding solutions to programming contests. The authors propose 'context-free grammars with counters" as a means of representing test case generators. These grammars are learned from natural language code documentation, if I understand correctly, and are the leveraged to generate test cases via sampling. Some experiments are presented.

I found the paper poorly written with many English mistakes. It is unclear if the proposed techniques generalize beyond programming contests. Can the authors please explain what is specific to the programming solutions that required the specific grammar?

The neural translation needs to be explained in more detail; it is unclear if it is new.

The use of "soundness" and "completeness" is misleading as the proposed techniques are neither sound nor complete.

**Strengths:**

The paper addresses machine learning techniques to help with test case generation, hence it can be considered in scope for this conference.

**Weaknesses:**

The paper is poorly written and the contribution is unclear. Thus I am not sure that the paper is ready for publication.
The authors should evaluate beyond programming solutions.

**Questions:**

1. what is specific to the programming solutions that required the specific grammar?

---

### Official Review · Reviewer_Cepj · 2023-10-31

**Soundness:** 1 poor
**Presentation:** 1 poor
**Contribution:** 1 poor
**Rating:** 1
**Confidence:** 4

**Summary:**

The paper aims to automatically generate test cases for competitive programming
from natural language. It translates problem descriptions from natural language
to a formal grammar, and then leverages a test generation method to generate
test cases.

**Strengths:**

The problem is interesting.

**Weaknesses:**

The paper is very very very confusing and poorly written. For example, the
definitions in page 3, $X_A$ which represents $X_I \times A$ (where the symbol
$A$ is first time use without any explanation!) -- how is that possible, using a
single subscript to denote so much information? Are not all the following terms,
e.g., $x_0 = (x_i, 0)$, ambiguous? Is $x_0$ a single tuple? Then, for which $x_i$?
Or it consists of multiple ones -- namely, a set, in which case, you should write it as a
set instead of a tuple? These definitions are simply not acceptable -- it does
not simplify but creates confusions and ambiguity!

Also, based on your definitions of $X_I$ and $X_A$, if I have a symbol, $X_B$ --
could you let me know it means $x_b \in X_B$ or $(x_i, b) \in X_B$? It is simply
impossible! As such, I cannot understand the paper as there are way too many
confusing terms and symbols.

**Questions:**

N/A

---

### Official Review · Reviewer_862x · 2023-11-01

**Soundness:** 2 fair
**Presentation:** 2 fair
**Contribution:** 2 fair
**Rating:** 3
**Confidence:** 3

**Summary:**

The paper presents a neural translation method to map natural language specifications of test cases to formal grammars, which are then used to sample test cases for software programs. A context-free grammar is defined to represent the necessities for the test cases. The focus is limited to competitive programming problem descriptions. Experiments are performed and compared to some baselines, both LLM-based and using fuzzing.

**Strengths:**

The generation of test cases is a relevant topic and the translation of natural language to actual source code can be meaningful approach. The paper exploits the popularity of language models and code-focused variants, such as CodeT5 and explores their applicability for this use case.

In the experimental evaluation, not only LLM-based techniques are considered, which is good. Mutation-based fuzzing is presented as an alternative from software testing.

**Weaknesses:**

The actual contribution of the paper seems limited; the main proposition is probably the specification of the context-free grammar with counters that suits the set of (semi-structured) problem descriptions and required test cases. Using LMs to sample to a specific grammar is nowadays a more-or-less common approach (e.g. it's even implemented in standard libraries for LLMs like llama.cpp) and even with the grammar output of the model, there is still substantial sampling (beam search) required to arrive at good test cases.

Mutation-based fuzzing is a relatively basic test generation technique, only slightly better than random inputs, and it does not seem to be a good representative of a strong baselines or even state-of-the-art techniques. Grammar-based fuzzing or search-based testing techniques are nowadays easily applicable and should be a more relevant baseline technique.
Related to this, the paper seems to be dismissive of the activities in the test case generation in the field of software engineering and software testing. Few papers from this field are cited, but there is an enormous body of work available that addresses the similar approach.

For the presented results, I'm having some trouble with the interpretation. In Table 3 some numbers are marked in bold, which do not actually seem to be the best results, e.g. CCFGT5/Beam(10) has 75.75 valid while ChatGPT and Contests/Generated have higher values. Same for the coverage, which by itself is not a meaningful metric, but still 100% is usually better than 88%. This is unclear and needs better explanation in Sec. 4.6.3.

In conclusion, the general problem is interesting, but the actual paper is very limited on the competitive programming problem descriptions, which are somehow semi-structured. The evaluation shows using a grammar can work better in this setting than few-shot learning with LLMs. I do not recommend to accept this paper at this stage.

**Questions:**

- How are the results in Table 3 to be interpreted?